# Primary blast wave protection in combat helmet design: A historical comparison between present day and World War I

**Joost Op 't Eynde⬤\*, Allen W. Yu⬤, Christopher P. Eckersley⬤, Cameron R. Bass**

Department of Biomedical Engineering, Duke University, Durham, North Carolina, United States of America

\* joost.opteynde@duke.edu

**Data Availability Statement:** All the data underlying the analysis can be found in a public repository hosted by the Duke Data Repository. All

## Abstract

Since World War I, helmets have been used to protect the head in warfare, designed primarily for protection against artillery shrapnel. More recently, helmet requirements have included ballistic and blunt trauma protection, but neurotrauma from primary blast has never been a key concern in helmet design. Only in recent years has the threat of direct blast wave impingement on the head–separate from penetrating trauma–been appreciated. This study compares the blast protective effect of historical (World War I) and current combat helmets, against each other and 'no helmet' or bare head, for realistic shock wave impingement on the helmet crown. Helmets included World War I variants from the United Kingdom/United States (Brodie), France (Adrian), Germany (Stahlhelm), and a current United States combat variant (Advanced Combat Helmet). Helmets were mounted on a dummy head and neck and aligned along the crown of the head with a cylindrical shock tube to simulate an overhead blast. Primary blast waves of different magnitudes were generated based on estimated blast conditions from historical shells. Peak reflected overpressure at the open end of the blast tube was compared to peak overpressure measured at several head locations. All helmets provided significant pressure attenuation compared to the no helmet case. The modern variant did not provide more pressure attenuation than the historical helmets, and some historical helmets performed better at certain measurement locations. The study demonstrates that both historical and current helmets have some primary blast protective capabilities, and that simple design features may improve these capabilities for future helmet systems.

## Introduction

'That men do not learn very much from the lessons of history is the most important of all the lessons that history has to teach.'—Aldous Huxley.

At the start of World War I (WWI) in July 1914, helmets were not part of the standard military equipment for any of the allied or central powers [1]. Most headwear consisted of cloth (e.g. French Kepi [2]) or leather (e.g. German Pickelhaube [3]) and did not offer the wearer any protection from blasts, shrapnel, or ballistic impacts. Multiple reports at the time estimated

46 files are avalible at https://doi.org/10.7924/r4r49m981.

**Funding:** We would like to gratefully acknowledge funding and support from the Josiah Charles Trent Memorial Foundation Endowment Fund at Duke University for directly funding this research. There is no particular grant associated with the work and funding was not provided to a particular member of the research group. The funders had no role in study design, data collection and analysis, decision to publish, or preparation of the manuscript.

**Competing interests:** The authors have declared that no competing interests exist.

that at the start of the war, over fifty per cent of fatalities occurred due to shrapnel or artillery shell fragments, often striking the head, for which steel helmets could be effective [1, 4].

In 1915, France was the first nation in WWI to equip soldiers with steel helmets, utilizing the M15 Adrian helmet, named after the design by General Adrian [5]. Inventor John L. Brodie addressed the British need for head protection in late 1915 with a helmet design aimed at shrapnel protection while focusing on ease of manufacturing [6]. Other nations also used the Brodie helmet, including the United States when they joined the war in late 1917 [7]. After extensive testing of Allied helmets, the Stahlhelm (translation: steel helmet) was rolled out to German soldiers at the start of 1916 [8].

These helmets were effective in their design to protect against artillery shell shrapnel [1]. Besides propelling shrapnel, exploding artillery shells also create a shock wave. The shock wave is referred to as primary blast, while the projectiles launched during an explosion are considered secondary blast. In WWI, the effects of these blast waves were experienced on a large scale for the first time in the combat theatre. Soldiers who experienced explosions in close vicinity were delivered to field hospitals despite having little to no signs of external trauma. British physician Charles Myers used the term 'Shell Shock' in 1915 to describe an array of symptoms experienced by soldiers after shell explosions [9], today believed to be potentially caused by a combination of traumatic brain injury (TBI) and psychological trauma [10].

Since the early 1900s, exploding artillery shells have been the largest cause of combat casualties in major conflicts [11]. In US wars since WWI, there has been an increasing trend towards greater number of casualties being caused by explosions, with one study reporting 78% of all injuries in the 2001–5 period of the conflict in Iraq being caused by explosions [12]. During the conflicts in Iraq and Afghanistan, over 65% of reports of TBI were associated with an explosion [13]. A 2008 study of US Army Infantry soldiers returning from deployment in Iraq found more than 15% of them suffered some form of mild traumatic brain injury (mTBI) [14]. In the past decade, there has been an increase in the awareness of long-term debilitating effects of primary blast mTBI, such as axonal injuries [15, 16]. Blast exposures causing minimal acute injuries might cause functional brain changes over time or with repeated exposures. Since most blast TBI is classified as 'mild,' there is an increasing demand for combat helmets to protect against these exposures.

Risk assessment of human blast injury to the pulmonary system was developed in the 1960s [17] and recently given a stronger experimental basis [18, 19]. The bulk of historical blast work implies that pulmonary tolerance is much lower than neurotrauma tolerance for blast [20–22]. This has been recently confirmed in direct comparative experimentation in rabbits [23]. In concert with these studies, researchers have recently developed risk functions for potential neurotrauma from blast [23, 24] that provide assessment tools for primary blast effects on the head. Initially perplexing, the incidence of blast pulmonary trauma following blast exposure in current military conflicts is quite low compared with blast neurotrauma [25], despite the difference in tolerance. This apparent contradiction is resolved by the almost universal use of body armor which dramatically increases the pulmonary tolerance of blast relative to that of the brain [26].

The battlefield conditions of WW1 on the Western Front provide similar blast conditions to those seen today. Battles fought from trenches in short spurts of unit advancement largely result in the helmeted head being exposed to the blast, while the torso is more distant or covered, decreasing the potential for blast pulmonary trauma. No current fielded helmet system has been specifically designed for blast protection, though careful studies suggest that modern helmets have a degree of blast protective effects [27–29].

This study compares the blast protective capabilities of principal military helmets from WWI combatants with a modern composite helmet. For the three historical helmets discussed

in this study, no record of primary blast evaluation was found in the scientific literature. The current study is, to our knowledge, the first to assess the protective capabilities of these historical combat helmets against blast. Brain injury due to primary blast was first recognized around the same time these helmets were being developed [30], and primary blast is now a generally recognized mechanism of injury to the brain. This study is an investigation into whether improvements have been made in combat helmet primary blast protection or if there is a lesson to be learned from these 100-year-old designs.

## Methods

### Helmets

Three authentic historical WWI infantry combat helmets including the original lining, were acquired for blast testing: an M15 (1915 model) Adrian Helmet used by the French Army (denoted FRC), an M1916 Stahlhelm used by the Imperial German Army (denoted GER), and an M1917 Brodie Helmet used by the U.S. Army (based on the M1915 British design and denoted AMR). The M1917 Brodie Helmet was manufactured by the Columbian Enameling and Stamping Company (Terre Haute, IN, USA). The Advanced Combat Helmet, the current combat helmet used by the U.S. Army, was included (size large, denoted ACH) for comparison to current combat helmets. A 'no helmet' bare head case was used as a control (denoted BAR).

The three WWI helmets are made of formed steel, and the Advanced Combat Helmet (ACH) has a fiber composite construction. The average wall thickness of each helmet was measured using electronic calipers (EC799, L. S. Starrett Company; Athol, MA, USA). The projected area for the top view of each of the helmets was calculated using ImageJ (NIH; Bethesda, MD, USA). Weight, wall thickness, and projected area of each of the helmets, and abbreviations used in this manuscript are described in Table 1. High resolution X-ray computed tomography images(Nikon XTH 225 ST, Nikon Inc.; Minato, Tokyo, Japan) were made of the historical helmets and coronal sections are displayed in Fig 1.

### Blast setup

Helmets were mounted on a Hybrid III® 50th percentile male dummy head (Humanetics; Farmington Hills, MI, USA) and affixed to the Hybrid III neck. Each helmet was secured around the chin and back of the dummy head (Fig 2) to prevent extraneous helmet motion during testing. Original buckles and leather straps were not used due to the degraded conditions that would not withstand the blast scenarios. For the ACH, original helmet straps were used. The ACH fit properly on the dummy head, covering the forehead while leaving one to two centimeters space above the eyes as described in the ACH operator's manual. The historical helmets all fit on top of the head, with the head held in the internal suspension without the crown of the head touching the helmet. Each helmet had both the external steel components and internal textile/leather components intact. The dummy head was faced downwards, and the center of the head was aligned with the open end of a cylindrical blast tube (schematic in Fig 3). This

**Table 1. Helmet abbreviations and properties.**

| Helmet | Abbreviation | Weight (kg) | Average thickness (mm) | Projected area from above (cm²) |
|---|---|---|---|---|
| Adrian M15 helmet | FRC | 0.67 | 0.75 | 439.9 |
| Stahlhelm M1916 | GER | 1.23 | 1.20 | 544.5 |
| Brodie M1917 helmet | AMR | 0.88 | 0.95 | 689.2 |
| Advanced Combat Helmet | ACH | 1.51 | 8.40 | 516.4 |
| Bare head | BAR | / | / | 327.6 |

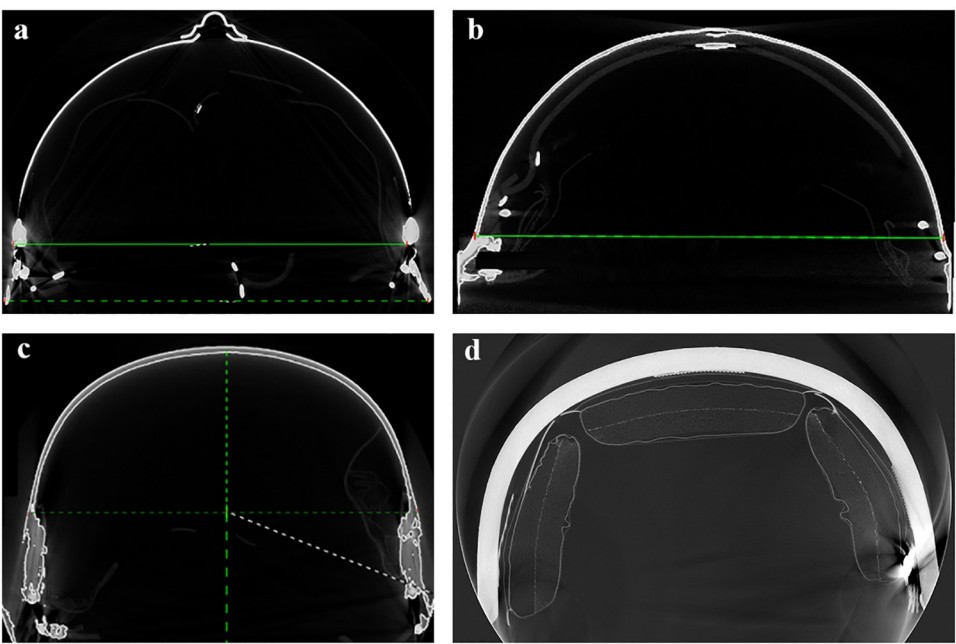

**Fig 1. Coronal CT sections of tested helmets.** (a) Adrian helmet, (b) Brodie helmet, (c) Stahlhelm, (d) Advanced Combat Helmet (ACH). The Adrian helmet (a) is the thinnest steel followed by the Brodie (b) and the Stahlhelm (c). The ACH (d) is made with a fiber composite material.

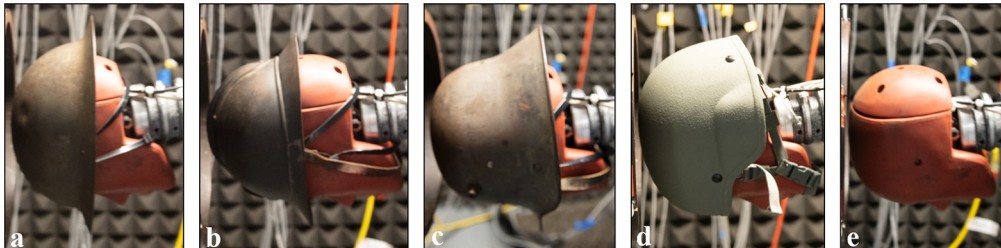

**Fig 2. Helmets on Hybrid III head in test setup.** (a) Brodie helmet, (b) Adrian helmet, (c) Stahlhelm, (d) Advanced Combat Helmet, (e) No helmet.

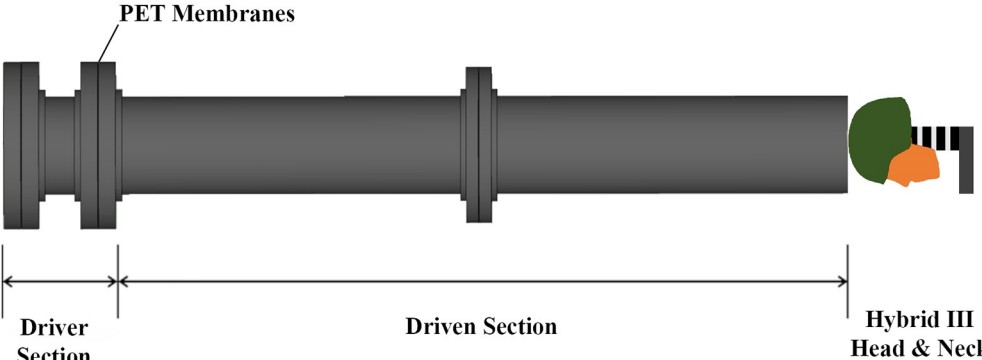

**Fig 3. Shock tube setup schematic.** Exposure is in the direction of the crown, simulating overhead blast that is typical exposure for personnel in trenches or prone on the ground.

orientation and blast exposure simulate an overhead blast scenario, as would have been common in trench warfare due to artillery shells exploding above trenches.

The top of the helmet was aligned with the end of the blast tube to minimize standoff distance. The blast tube has a diameter of 305 mm and consists of a driver section (305 mm length), where helium gas is compressed, and a driven section (3.05 m length). The driver and driven section are separated by a diaphragm consisting of a number of polyethylene terephthalate (PET) membranes. High pressure helium is released into the driver section until the PET diaphragm bursts, allowing a shock wave to travel down the driven section of the blast tube. The 10:1 driven length to driver length ratio allows the shockwave to develop a uniform shock front, with approximate equal pressure across the tube section [31]. In a previous study [31], it was shown that testing outside the blast tube is appropriate as long as standoff distance is minimized.

The helmets were exposed to shock waves at three separate blast intensities by varying the thickness of the bursting diaphragm with PET membranes: two membranes of 0.254 mm thickness (total thickness: 0.508 mm), nine (2.286 mm), and twelve (3.048 mm). These choices for membrane thickness and resulting shock intensity were made to represent historical blast exposure (see blast simulation) and approximate blast levels corresponding to 50% risk for respectively mild meningeal bleeding, moderate meningeal bleeding, and severe meningeal bleeding based on bare head ferret brain blast data from Rafaels et al. [24]. In total, forty-six blast tests were performed for this study, detailed in Table 2. All two-membrane tests were performed first in the order ACH-FRC-AMR-GER-BAR, followed by all nine-membrane tests in the order BAR-GER-AMR-FRC-ACH, and finally all twelve-membrane tests in the order ACH-AMR-FRC-GER-BAR. For each helmet all tests at a specific number of membranes were performed consecutively.

## Data acquisition

Blast overpressure was measured using three pressure transducers (Endevco 8530B; San Juan Capistrano, CA, USA) at the exit of the shock tube, inside the tube wall, incident to the direction of the wave. One transducer was positioned at the top of the tube, with the two others symmetrically positioned 120˚ clockwise and counter-clockwise from the position of the first. In addition to tube pressure measurements, five pressure transducers were inserted in the Hybrid III dummy head. Oriented radially outward, transducers were located at the crown, forehead, right ear, left eye, and back of the head. The pressure measurement locations on the Hybrid III Head are indicated on Fig 4. A three-axis load cell (Model 2564, Robert A. Denton, Inc; Rochester Hills, MI, USA) was mounted between the Hybrid III Head and Neck, to examine neck forces. Neck load cell data are not reported in the current study. Pressure and forces were sampled at 200 kHz using a meDAQ® (Hi-Techniques, Inc.; Madison, WI, USA) data acquisition system. Pressure traces were demeaned to an ambient pressure of zero but were not filtered. An example pressure trace for each helmet at the highest tested pressure amplitude is shown in Fig 5. High speed video of each blast was collected at 8000 fps using a Phantom® V711 camera (Vision Research; Wayne, NJ, USA).

**Table 2. Number of blast tests performed for each case, 46 in total.**

| Helmet | Number (thickness) of PET burst membranes | | |
|:---:|:---:|:---:|:---:|
| | 2 (0.508 mm) | 9 (2.286 mm) | 12 (3.048 mm) |
| BAR | 4 | 4 | 3 |
| ACH | 3 | 3 | 3 |
| GER | 3 | 4 | 2 |
| AMR | 3 | 3 | 2 |
| FRC | 3 | 3 | 2 |

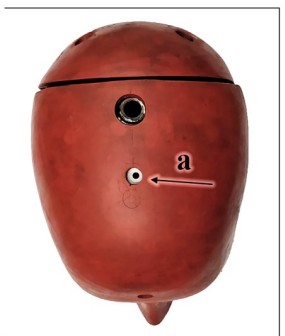 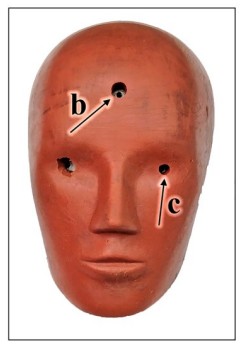 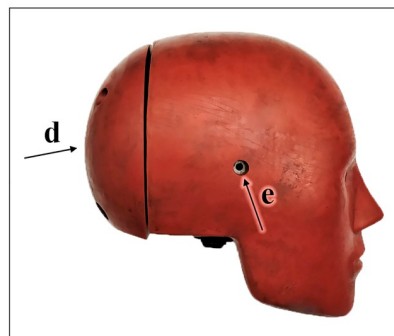

**Fig 4. Views of Hybrid III head with pressure sensor locations indicated.** (a) crown, (b) forehead, (c) eye, (d) back head, and (e) ear.

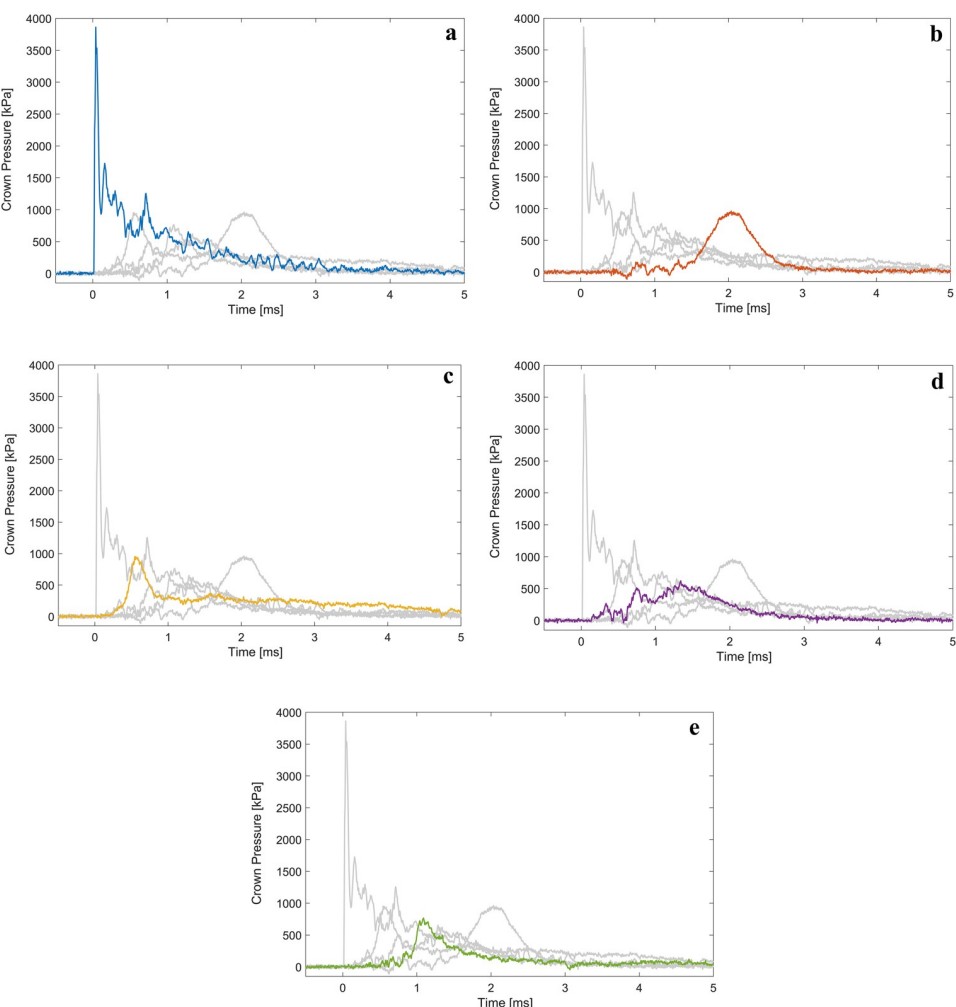

**Fig 5. Representative pressure traces.** Measured at the crown of the dummy head, for 12 membrane (3.048 mm PET) tests of the (a) Bare Head, (b) Advanced Combat Helmet, (c) Brodie Helmet, (d) Adrian Helmet, and (e) Stahlhelm.

## Pressure conversion

To compare pressure measured by 'side-on' transducers in the wall of the blast tube to pressure measured by the 'face-on' transducer at the crown of the head, Rankine-Hugoniot relationships were used to convert the incident pressure to reflected pressure (Eq 1) [32].

$$P_{refl} = 2P_{inc} \left( \frac{7P_{atm} + 4P_{inc}}{7P_{atm} + P_{inc}} \right) \tag{1}$$

In Eq 1, $P_{refl}$ is the reflected pressure (gauge), $P_{atm}$ is the atmospheric pressure, set to 101.325 kPa, and $P_{inc}$ is the measured incident pressure (gauge).

## Injury risk curves

Injury risk curves from Rafaels et al. [24] were used to provide a brain injury risk value for the measured crown overpressure. Meningeal bleeding risk curves were obtained from scaled ferret blast brain experiments, with mild, moderate and severe bleeding defined by brain surface area covered by hemorrhaging ($< 3\%$, $3–10\%$, and $> 10\%$ respectively). The duration of blast in these experiments was scaled to a reference mass of a 70 kg human using cube root of body mass scaling. Using Eq 1, the risk curves were converted from incident to reflected pressure.

Risk curves from Richmond et al. [33] were used in conjunction with pressure measurements at the ear location to estimate the risk for eardrum rupture. The measurements at the ear were recorded with an incident orientation to the shock wave. Pressure-time curves for the 50% injury risk of three eardrum rupture levels were defined: minor (minor slits, linear disruption of drum fibers), moderate (large tears, multiple small holes or tears), and major (total disruption, large flaps of the drum).

## Blast simulation

Blast conditions used in this study were compared to WWI Germany artillery shell explosions to determine comparable exposure ranges. Details on German artillery shells are shown in Table 3 [34]. These shells represent most of the German artillery fired on the Western Front [35]. The distance from the charge to experience a blast similar to the tested conditions was calculated for each of these shells using ConWep (U.S. Army Corps of Engineers, Protective Design Center, Vicksburg, MS). Tested blast conditions were binned into 3 severity groups based on PET membrane thickness used to generate the blast (Table 4). Mean bare head crown pressure and positive phase duration were used as a representative blast for each severity level. The results of the simulations are displayed in Fig 6, showing at what range the tested blast conditions would compare to WWI German artillery shell explosions.

## Statistical analysis

JMP$^{\circledR}$ Pro 13.0.0 (SAS Institute Inc.; Cary, NC, USA) was used for statistical analysis. A general linear model (GLM) was constructed with peak crown pressure as the outcome variable.

**Table 3. German artillery rounds used in WWI [34].**

| German artillery | Shell weight (kg) | Explosive charge (kg-TNT) | Rounds fired (million) |
|---|---|---|---|
| 77 mm FK | 6.5 | 0.4 | 156 |
| 105 mm FH | 15 | 1.5 | 67 |
| 150 mm FH/K | 38.6 | 5.6 | 42 |
| 210 mm Mörser | 114.5 | 11.6 | 7 |

**Table 4. Severity bins for bare head blasting conditions.**

| Severity | PET membranes (thickness, mm) | Peak crown pressure (kPa +/- SD) | Positive phase duration (ms +/- SD) |
|---|---|---|---|
| Low | 2 (0.508) | 880 +/- 91 | 1.47 +/- 0.11 |
| Medium | 9 (2.286) | 3558 +/- 224 | 3.25 +/- 0.07 |
| High | 12 (3.048) | 4521 +/- 488 | 3.66 +/- 0.13 |

SD, standard deviation

Helmet type (no helmet, ACH, Brodie, Adrian, Stahlhelm) and measurement location (crown, forehead, back head, eye, ear) were included as factors with measured peak tube pressure as a covariate. All interaction terms were found to be significant (p<0.0001), so a subdivision was made to consider each measurement location separately. The interaction between helmet type and peak tube pressure was found to be significant (p<0.0001) for each measurement location and was included in the model (i.e. the slope of the linear regression between tube pressure and crown pressure differs by helmet). Pairwise comparisons between the helmets were done by examining the significance of the interaction term between helmet type and tube pressure when including only the compared helmet conditions in the model. If the interaction term was not significant, it was removed from the model to examine an overall helmet effect when slopes are equal.

## Results

In total, forty-six blast tests were performed for the five helmets over three blast conditions described in Table 2. After observing deformations that may affect the structural integrity in the Brodie and Adrian helmet (Fig 7) at the highest tested peak pressures, it was decided to keep the number of 3.048 mm PET tests at two for the historical helmets. For all other

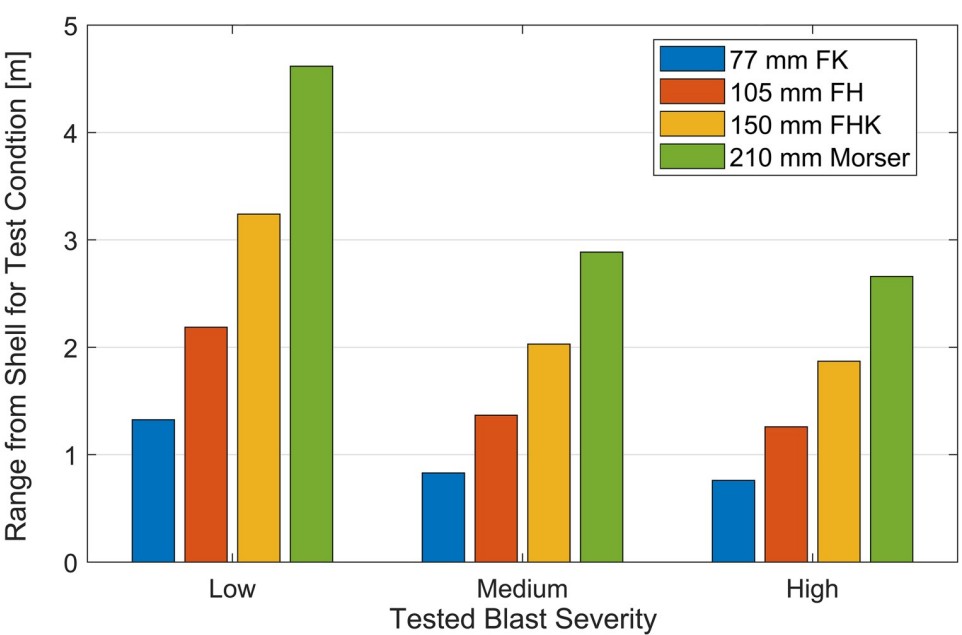

**Fig 6. Equivalent distance to WWI shell explosions.** Distance from a WWI German shell explosion to experience blast conditions tested in this study. Pressure measurements from bare head testing conditions were used to calculate these distances.

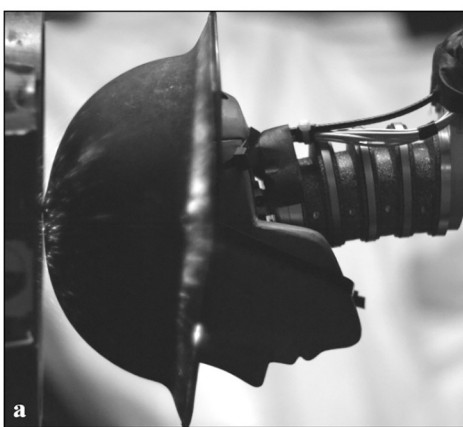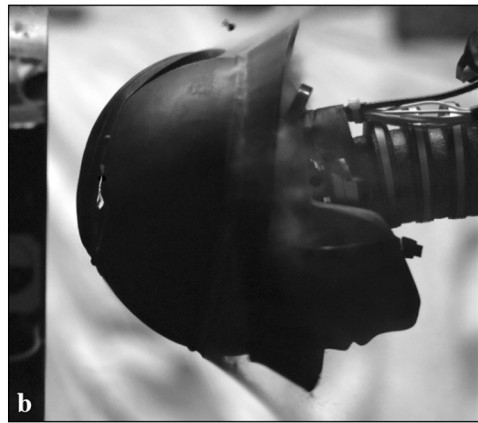

**Fig 7. High speed frames of blast impact on the head.** Frames from high speed video recording of high intensity blast tests immediately after blast wave impingement on the (a) Brodie helmet and (b) Adrian helmet.

exposures, helmets experienced minimal deformation and no evidence of degradation by repeated blast exposure was observed. Blunt impact of helmet deformation on the head was not assessed in this study.

Peak tube pressure, helmet type, and their interaction were each found to have a statistically significant effect on the crown pressure for each measurement location ($p < 0.001$). Statistical significance of the interaction term between helmet type and tube pressure justifies the use of different regression slopes for the different helmet types.

At the crown of the head, the interaction term differed significantly between the bare head and all helmets ($p < 0.0001$), with higher crown pressures for the bare head. There was also a significant difference in slope between the ACH and the French Adrian helmet ($p < 0.0001$). When removing the interaction term, the Adrian Helmet results showed a significant difference in pressure compared to the British/American Brodie helmet and the German Stahlhelm ($p < 0.01$). The Adrian helmet resulted in lower crown pressures than all other cases. The ACH, Stahlhelm (GER), and Brodie helmet (AMR) were not found to be significantly different from each other ($p > 0.05$). The results of the general linear model for the crown measurement location are shown in Fig 8. Some of the regression lines for the helmet blast results do not pass through the origin, suggesting that the peak pressure attenuation provided by helmets might be nonlinear at lower blast pressures.

Besides lower peak pressures, the crown pressure traces also showed a more gradual loading rather than a near instantaneous shock front when assessing the helmeted case compared to the bare head (Fig 5). Video analysis of the blast event showed that the delay in pressure rises seen in the figure corresponds to the helmet moving in the suspension towards the head, and the peak pressure time roughly corresponds to the maximum compression of the helmet suspension. However, because of the short durations ($< 2$ ms) of these pressure peaks, the use of a blast injury criterion was deemed appropriate.

The blast tests carried out at different amplitudes were found to be in the 50% risk range for mild, moderate and severe meningeal bleeding for crown pressure on the bare head (Fig 9) based on the scaled ferret risk curves (section 2.5) [24]. Wearing a helmet was associated with a decrease in bleeding risk. This shows that the performed tests simulate realistic exposures where wearing a helmet might change physiological outcomes in the brain. In Fig 10, the 50% moderate meningeal bleeding case for the bare head is compared to the helmet results at that level. For the same blast conditions, risk of moderate bleeding is lower than 10% in all helmets, and close to 1% for the Adrian helmet.

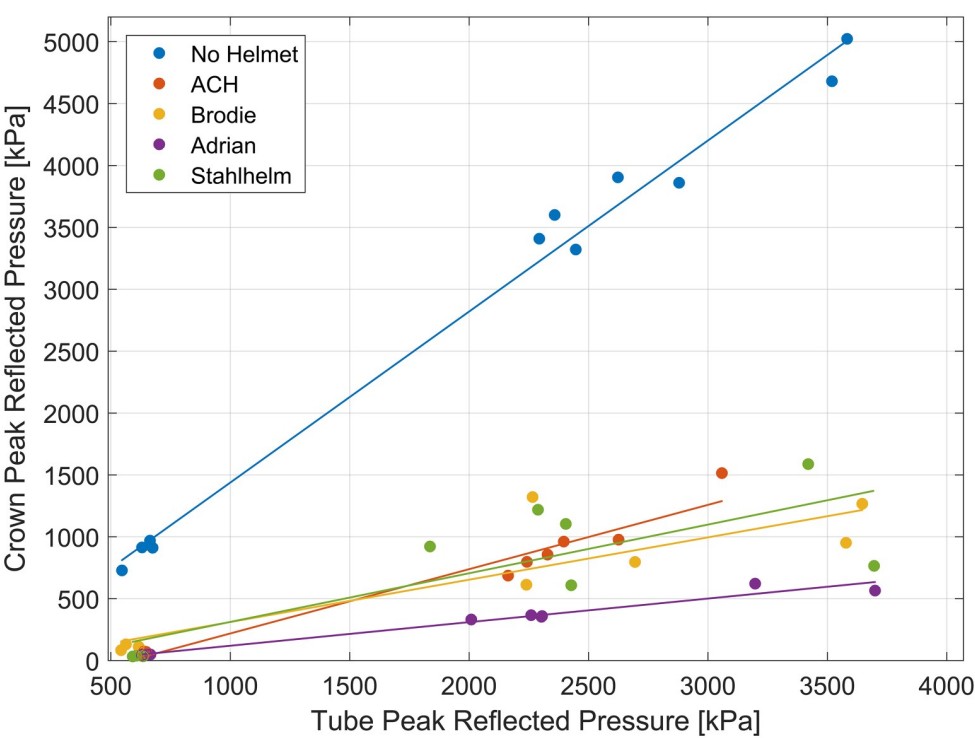

**Fig 8. Higher crown pressures seen without helmet, lower with Adrian helmet.** Measured tube and crown pressure for each test performed, and the linear regression fit for each helmet type. Bare head crown pressure is higher than all helmets (p<0.0001), and the French Adrian helmet crown pressure is lower than all other helmets (p<0.01).

At the forehead measurement location (Fig 11a), there was a significant difference in slopes between the bare head and all helmets (p<0.0001), and the Stahlhelm was different from the ACH and Brodie helmet (p<0.05) with higher pressures. No additional differences were found when the interaction term was removed (p>0.05).

For the back of the head (Fig 11b), there was again a significant difference in slopes between the bare head and all helmets (p<0.0001). When removing the interaction term to consider equal slopes, the Stahlhelm was found to differ from the Brodie helmet and ACH (p<0.01), and the Adrian helmet also differed from the Brodie helmet and ACH (p<0.005).

For pressure measured at the left eye (Fig 11c), the bare head and the ACH both differ in slopes compared all other helmets (p<0.0001) with higher pressures. The Adrian helmet slope differs significantly from the Brodie helmet (p<0.001). With equal slopes, the bare head has a significantly higher pressure than the ACH (p<0.0001).

At the right ear measurement location (Fig 11d), the bare head had a steeper slope than all helmets (p<0.0001). Comparing with equal slopes, the Adrian helmet had a higher pressure than the Brodie helmet and ACH (p<0.01), and the Stahlhelm (p<0.05).

The ear pressures for the bare head condition exceeded 50% major rupture risk [33] for all tested severity levels (Fig 12). Rupture risk was reduced in all helmeted conditions, with less than 50% risk for a minor rupture at the low severity levels for all helmets except the Adrian helmet and 50% moderate to 50% major rupture risk at the medium and high severity levels.

## Discussion

Blast exposure to the bare head was more severe than any helmeted test for every blast intensity and at every measurement location. The bare head experienced three to five times higher peak

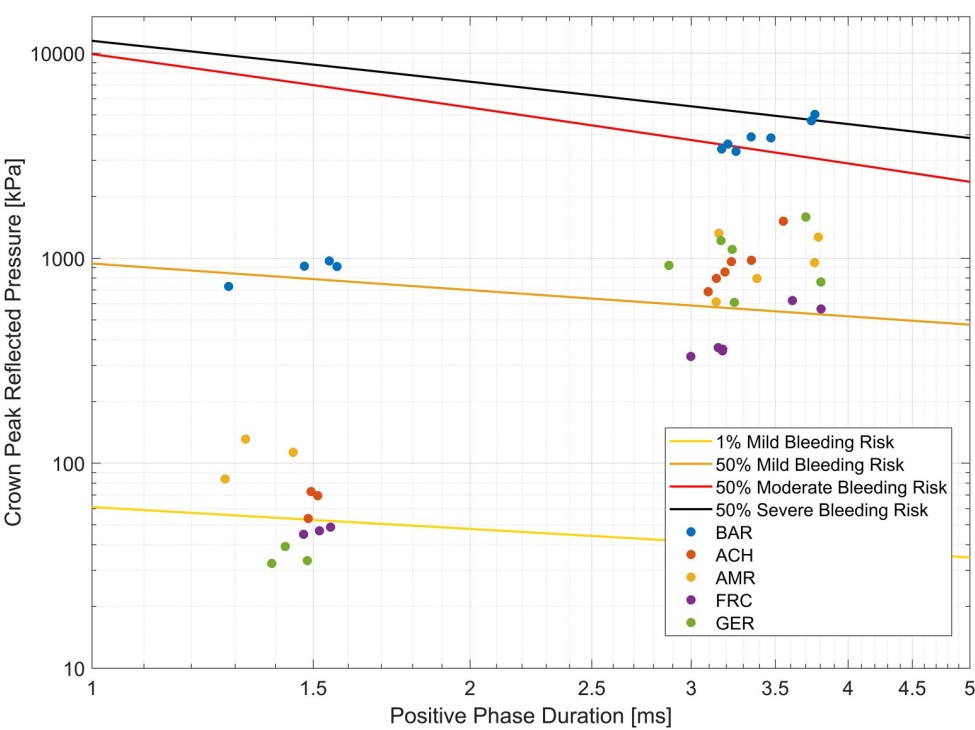

**Fig 9. Helmets reduce the risk of meningeal bleeding.** Tested blast conditions plotted on brain blast meningeal bleeding risk curves from Rafaels et al. [24]. Bare head testing conditions are roughly situated in the 50% mild, moderate, and severe meningeal bleed risk range, whereas the bleeding risk for helmeted tests is much lower (see Fig 10 for a more detailed example).

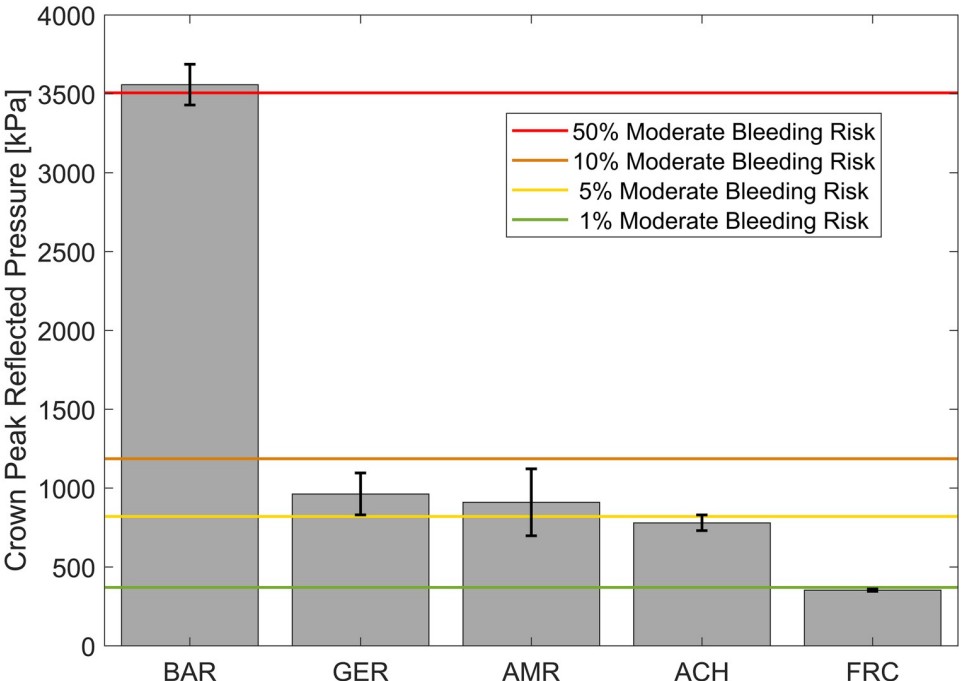

**Fig 10. Risk reduction for moderate bleeding in helmeted cases.** Mean peak pressure measured at the crown of the head for moderate severity blasts (Table 4), including standard error indicated on the bars. For approximately a 50% moderate bleeding risk in the bare head scenario, moderate bleeding risks for all helmets is more than 5x lower for the same testing condition.

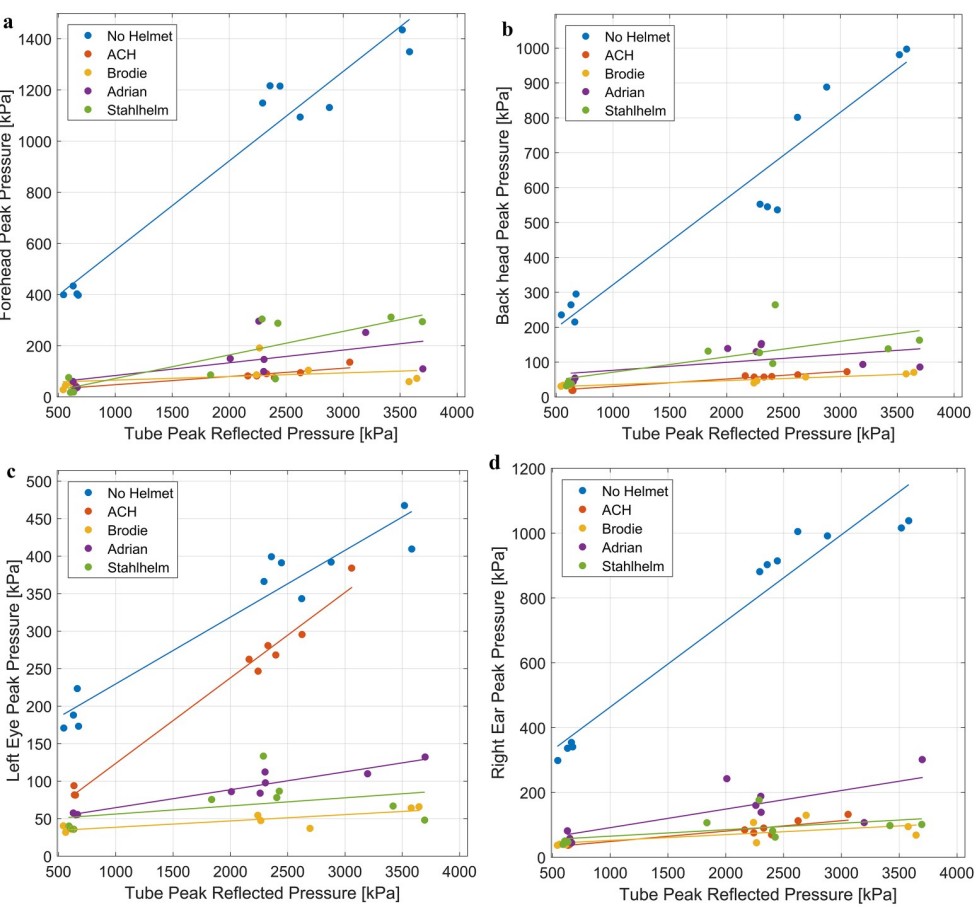

**Fig 11. Helmets reduce blast overpressure at all sensor locations on the head.** Linear regression fit for each helmet for peak pressure measured in the blast tube and peak pressure measured at each location on the dummy head (Fig 4). (**a**) Forehead: Bare head higher than all helmets (p<0.0001), and Stahlhelm higher than ACH and Brodie (p<0.05). (**b**) Back of the head: Bare head higher than all helmets (p<0.0001), Stahlhelm higher than ACH and Brodie (p<0.01), and Adrian higher than ACH and Brodie (p<0.005). (**c**) Eye: Bare head higher than all helmets (p<0.0001), ACH higher than all other helmets (p<0.0001), and Adrian higher than Brodie (p<0.001). (**d**) Ear: Bare head higher than all helmets (p<0.0001), Adrian higher than Brodie, ACH (p<0.01), and Stahlhelm (p<0.05).

pressures (Fig 8) at the crown of the head (at similar positive phase durations), which corresponds to higher risk of meningeal bleeding and other potential brain injuries [24]. Helmets provided more shock wave attenuation at lower pressure levels than at higher pressure levels (Fig 8), suggesting that helmets might play an especially important role in protection against mild primary blast induced brain trauma. The effect of wearing a helmet, especially for short positive phase durations (0.5–5 ms), is a significant reduction in risk of blast brain injury at the crown of the head for overhead blast scenarios. In other orientations, blast wave measurements are complicated by the difference between reflective (measured with pressure gauges oriented parallel to the direction of the blast) and incident (measured with pressure gauges oriented perpendicular to the direction of the blast) pressures, leading to conflicted reports of helmets possibly increasing the risk of primary blast injury [36–42]. This risk has to be carefully evaluated because reflected pressure measurements can be two to eight times greater than incident pressure measurements for the same blast scenario [32].

An interesting result from these experiments is the blast protective effect provided by the French Adrian helmet, which had a lower crown pressure than all other helmets, despite being

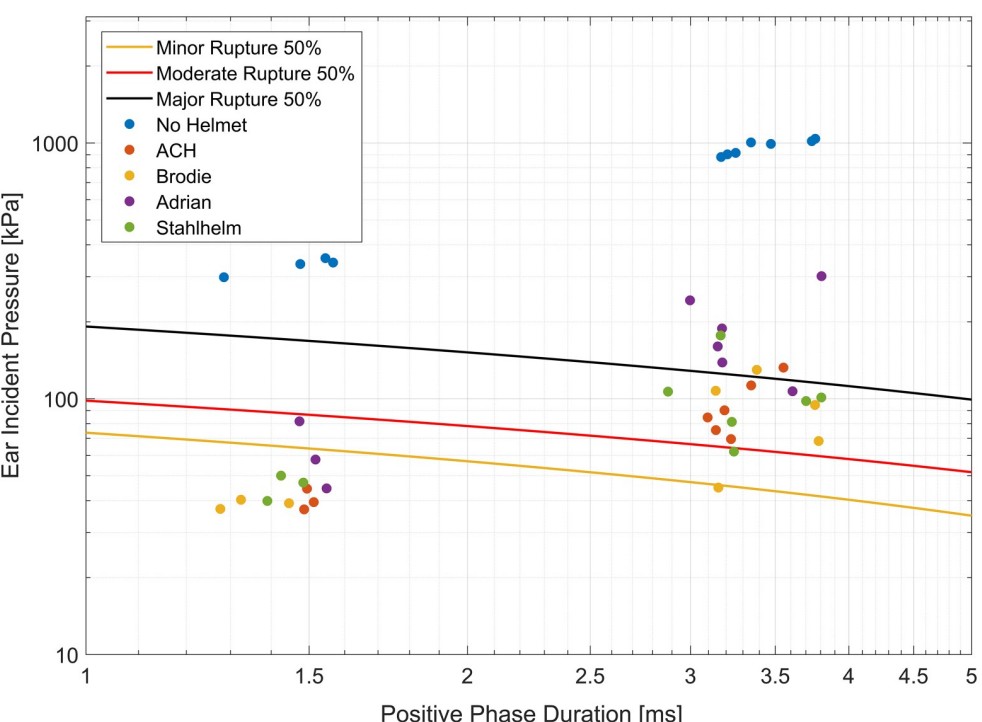

**Fig 12. Helmets reduce the risk of eardrum rupture.** Measured peak incident pressures at the ear and eardrum rupture risk curves from Richmond et al. [33]. Risk of eardrum rupture is lower in all helmeted cases compared to the bare head, and is higher in the French Adrian helmet compared to other tested helmets.

manufactured using similar materials as the Stahlhelm and Brodie Helmet, with a thinner helmet wall (Table 1). This result might stem from the deflector crest along the midline of the helmet (Fig 1a). Specifically added with overhead shrapnel in mind [43], this feature of the helmet could deflect the shock wave off to the side of the head, rather than allow shockwave impingement onto a more planar surface seen in the other helmets. The crest also provides an added first layer for shock wave reflection before reflecting a second time off the helmet itself. The crown pressure sensor used in the measurements was located under the deflector crest and may have experienced a decreased peak pressure because of this. Further studies are needed to see if surface geometry manipulation or helmet attachments may augment the protective capabilities of helmets against blast exposure.

Peak pressures measured in locations other than the crown of the head were much lower because of measurement at an orientation incident to the blast wave and being partly or completely covered by the helmets. In these locations, the Adrian helmet did not provide the same protective advantage seen at the crown. Pressure attenuation was seemingly determined by the width of the brim and/or coverage of the helmet (Fig 2). At the ear, the small brim and limited coverage of the Adrian helmet resulted in higher pressures than the other helmets (Fig 11d), with a corresponding increased risk in eardrum damage (Fig 12). The ACH, without a brim as seen in the historical helmets, had increased pressures at the eye (Fig 11c) but provided similar protection at the other measurement locations.

While ballistic protection provided by helmets has increased significantly since WWI and saved many lives [39], the results found here suggest that the ACH did not perform quantitatively or qualitatively better than the historical helmets, and performed worse than the Adrian helmet for overhead primary blast at the crown of the head. On the other hand, while ballistic

protection has been an active focus in combat helmets design, protection from primary blast has not been an important design element [39], and the level of protection from primary blast from all of the helmets tested is large compared with the bare head. One of the reasons for this is that the mechanism for blast protection was poorly understood for the first sixty years following WWI. While the exact injury mechanism for primary blast is still unknown, the scientific community (cf. Cooper, 1991) [44] identified acoustic impedance as one important protection mechanism against blast waves.

The acoustic impedance protection mechanism against blast trauma is different than against ballistic trauma. An ideal protection against ballistic impacts can locally absorb high energy impacts without failure or excessive deformation by distributing the energy through the material [45]. Desirable materials have high strength, high modulus, and a high local speed of sound. Protection from primary blast waves can be obtained by attenuating the blast wave using an acoustic impedance mismatch at an interface the wave is travelling through. An increased difference in acoustic impedance causes a higher proportion of the blast wave to be reflected, rather than penetrate into the body where it causes local stresses and tissue damage [44]. The reflection coefficient $R$ can be calculated from Eq 2.

$$R = \frac{Z_{helmet} - Z_{air}}{Z_{helmet} + Z_{air}} \tag{2}$$

In Eq 2, $Z_{helmet}$ is the acoustic impedance of the helmet and $Z_{air}$ is the acoustic impedance of the air. Acoustic impedance $Z$ is calculated as the product of speed of sound in the material and density of the material. Ideal materials have a high local speed of sound and a high density. Steel has a greater acoustic impedance ($\sim 38 \cdot 10^6$ Pa·s/m$^3$ for hardened manganese steel [46], used in WWI helmets) than composite fibers ($\sim 12 \cdot 10^6$ Pa·s/m$^3$ for Kevlar$^{\circledR}$ 129, used in ACH [47]), but since both impedances are orders of magnitude higher than air ($\sim 440$ Pa·s/m$^3$), reflection will be relatively similar (R = 0.999977 for steel and R = 0.999927 for Kevlar$^{\circledR}$ 129). This explains the similar results for the ACH, Brodie helmet, and Stahlhelm. Many helmet and body armor materials have properties that are desirable for both ballistic and blast trauma. Because a shock wave reflection occurs at every interface where there is an acoustic impedance mismatch, primary blast protection can be improved by using multi-layered configurations of high and low acoustic impedance, with each layer reflecting a proportion of the penetrating wave. Not every layer of material will be beneficial to blast wave protection, and if a material has an acoustic impedance in between two neighboring materials, it will enhance blast wave penetration. The layered structure of the ACH might contribute to its blast protection, but future studies are needed to evaluate the effect of a layered structure.

Helmet wall thickness improves ballistic protection by providing higher strength and energy absorption, but it doesn't affect blast protection much since reflection only occurs at interfaces. While the Adrian helmet provided superior blast protection at the crown of the head for overhead blast in this study, Dean [1] noted that the ballistic protection it provided was less than both the Brodie helmet and Stahlhelm.

One of the limitations of this study is that only an overhead blast scenario was examined. While this would be an accurate approximation of blasts in trench warfare as in WWI or air bombings of soldiers in the field during major unit action, it would not be as applicable to other cases such as improvised explosive devices (IEDs) used as roadside bombs, a significant cause of injury and death in conflicts in Iraq and Afghanistan [48]. The current study evaluated primary blast protection without considering reflective surfaces. In combat scenarios, reflection of a blast wave off of a surface can change outcomes considerably [49], such as when a soldier lies on the ground with the crown of the head towards the blast, or is confined within a

trench. Another limitation is that the historical helmets tested are over one hundred years old, and their material properties might not be the same as they were originally. While properties of steel are relatively stable, the helmet linings may have degraded. However, there is no guarantee that replicas would be identical copies of the original either, so this study stays as true to the original helmets as possible. Finally, this study did not include the potential from blunt neurotrauma from impacts of the helmet on the head following acceleration from the transiting shock overpressure. This effect may be large with blasts that had larger positive phase duration and larger impulse than for the shells considered in this study.

## Conclusions

Interestingly, though primary blast protection was not a design objective, both historical and modern combat helmets provide primary blast protection. Tested modern helmets provide similar protection to historical ones in an overhead blast scenario. All tested helmets provided significant protection against primary blast brain injury compared to a bare head scenario. This protection substantially decreased the potential for primary blast neurotrauma from typical World War I artillery threat equivalents based on available injury criteria. While the helmets also provided protection against eardrum rupture based on current eardrum injury risk assessments, the resulting pressures were still injurious even with the helmets with extended brims. Major improvements made in helmet technology to increase ballistic protection do not provide the same increase in blast protection. At certain measurement locations, some historical helmets provided more blast attenuation than the modern helmet even though the modern helmets based on modern fiber composites are far more protective from typical ballistic threats. Specifically, the French 1915 Adrian helmet produced a lower peak pressure at the crown of the head compared to the Advanced Combat Helmet and the other historical helmets. These results show that there is considerable overlap in materials that have good qualities for ballistic and blast protection, but the protection mechanisms are different. Protection against primary blast focuses largely on impedance mismatches that reduce the amplitude of the transmitted waves to the head. The introduction of steel helmets during World War I reduced the toll of both blast and ballistics injuries at the front. In the future, helmet protection against primary blast might be improved by material choice, multiple material layers with different acoustic impedance, or the geometry of the helmet.

## Acknowledgments

The authors gratefully acknowledge the Department of Biomedical Engineering of the Pratt School of Engineering at Duke University for their support in this study.

## Author Contributions

**Conceptualization:** Joost Op 't Eynde, Allen W. Yu, Christopher P. Eckersley, Cameron R. Bass.

**Data curation:** Joost Op 't Eynde, Allen W. Yu, Christopher P. Eckersley.

**Formal analysis:** Joost Op 't Eynde.

**Funding acquisition:** Cameron R. Bass.

**Investigation:** Joost Op 't Eynde, Allen W. Yu, Christopher P. Eckersley.

**Methodology:** Joost Op 't Eynde, Allen W. Yu, Christopher P. Eckersley, Cameron R. Bass.

**Project administration:** Cameron R. Bass.

**Resources:** Allen W. Yu, Cameron R. Bass.

**Software:** Joost Op 't Eynde.

**Supervision:** Cameron R. Bass.

**Visualization:** Joost Op 't Eynde.

**Writing – original draft:** Joost Op 't Eynde.

**Writing – review & editing:** Joost Op 't Eynde, Allen W. Yu, Christopher P. Eckersley, Cameron R. Bass.

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
