## [Decision Letter · Decision Letter 0]

6 Nov 2019

PONE-D-19-23509

Primary blast wave protection in combat helmet design: a historical comparison between present day and World War I

PLOS ONE

Dear Mr. Op 't Eynde,

Thank you for submitting your manuscript to PLOS ONE. After careful consideration, we feel that it has merit but does not fully meet PLOS ONE’s publication criteria as it currently stands. Therefore, we invite you to submit a revised version of the manuscript that addresses the points raised during the review process.

We would appreciate receiving your revised manuscript by Dec 21 2019 11:59PM. To enhance the reproducibility of your results, we recommend that if applicable you deposit your laboratory protocols in protocols.io, where a protocol can be assigned its own identifier (DOI) such that it can be cited independently in the future. For instructions see: http://journals.plos.org/plosone/s/submission-guidelines#loc-laboratory-protocols

We look forward to receiving your revised manuscript.

Kind regards,

David Zonies

Academic Editor

PLOS ONE

Journal Requirements:

Reviewers' comments:

Reviewer's Responses to Questions

**Comments to the Author**

1. Is the manuscript technically sound, and do the data support the conclusions?

Reviewer #1: Yes

Reviewer #2: Yes

2. Has the statistical analysis been performed appropriately and rigorously? 

Reviewer #1: I Don't Know

Reviewer #2: Yes

3. Have the authors made all data underlying the findings in their manuscript fully available?

Reviewer #1: Yes

Reviewer #2: Yes

4. Is the manuscript presented in an intelligible fashion and written in standard English?

Reviewer #1: Yes

Reviewer #2: Yes

5. Review Comments to the Author

Reviewer #1: The authors present interesting data relevant to the common clinical problem of traumatic brain injury. They describe the historical context of the military helmet and its original intent to protect against ballistic injury. The authors then discuss the problem of primary blast injury. They describe how helmets were not designed for protection against primary blast injury. Helmet design has changed over time but it is unclear if these design changes have provided protection against primary blast injury. To test the effectiveness of modern and historical helmets, the authors used a model to simulate an overhead blast. They measured pressures at multiple head locations to assess the pressure attenuation provided by 3 historical and 1 modern helmet compared to no helmet. This was done at 3 different simulated blast intensities. The authors found that all helmets provided a significant pressure attenuation compared to no helmet. The modern design did not show a significant benefit over historic designs. This is interesting data that could benefit future improvements in protection against traumatic brain injury. This data is limited in being related only to an overhead blast as recognized by the authors. As helmets will need to provide protection from blasts from multiple angles, this data does not definitely say that any design is superior. Regardless, this is interesting information that could stimulate further efforts to improve the protective equipment mitigating the risk of traumatic brain injury.

Reviewer #2: Thank you for the opportunity to review this interesting analysis on helmet design and the protection they afford from primary blast injury. I have a few comments and questions for the authors.

1. The introduction was very interesting from a historic perspective; however, I think it could be condensed by about 30 to 50% while still retaining the essential historic information.

2. You should include more clinical justification for spending this much effort analyzing primary blast (e.g. expand on your references 14 and 15). As you indicate, most such primary blast injuries result in so-called "mild" TBI as there is often no overt injury (e.g. subdural or penetrating brain injury). Some very interesting work from USUHS by MacDonald and Brody, et al. has demonstrated changes on MRI in such patients (PMID: 21631321). There is growing recognition that "mild TBI" results in significant morbidity over time. This should be more fully enumerated to justify this work and future efforts to improve helmet design such that it prevents both penetration (secondary) as well as primary blast injuries.

3. Did the helmets lose protective capacity with repeated blast exposure?

4. The underwash argument sounds very similar to the explanation some unformed individuals give for why seatbelts should not be worn--i.e. if I wear a seatbelt and am involved in a motor vehicle crash, I could be trapped in the wreckage. This has been shown to be patently false. Were you able to assess the validity of this underwash argument? Or did your methods preclude such an assessment. If it is the former, please enumerate. If the latter, consider deleting mention of "underwash."

5. I disagree with the first sentence in your conclusions based on your introduction. It seems that primary blast protection was at least part of the motivation for designing helmets, at least in World War I. This may be less true today. Please clarify.

Just a few minor grammatical comments:

1. Avoid the use of contractions (e.g. lines 433 and 513).

2. Sentence 425 does not make sense. "Helmets provided...."

3. Consider deleting the preamble "The authors want to emphasize...." in sentence 428.

6. PLOS authors have the option to publish the peer review history of their article (what does this mean?). If published, this will include your full peer review and any attached files.

Reviewer #1: No

Reviewer #2: Yes: Jeremy W. Cannon, MD, SM, FACS

---

## [Author Response · Author response to Decision Letter 0]

21 Dec 2019

Q: The introduction was very interesting from a historic perspective; however, I think it could be condensed by about 30 to 50% while still retaining the essential historic information.

A: Condensed historical section of the introduction. Overall introduction length reduced from 1162 words to 874 words (including new additions). 

Q: You should include more clinical justification for spending this much effort analyzing primary blast (e.g. expand on your references 14 and 15). As you indicate, most such primary blast injuries result in so-called "mild" TBI as there is often no overt injury (e.g. subdural or penetrating brain injury). Some very interesting work from USUHS by MacDonald and Brody, et al. has demonstrated changes on MRI in such patients (PMID: 21631321). There is growing recognition that "mild TBI" results in significant morbidity over time. This should be more fully enumerated to justify this work and future efforts to improve helmet design such that it prevents both penetration (secondary) as well as primary blast injuries.

A: Included 

“In the past decade, there has been an increase in the awareness of long-term debilitating effects of primary blast mTBI, such as axonal injuries [16, 17]. Blast exposures causing minimal acute injuries might cause functional brain changes over time or with repeated exposures. Since most blast TBI is classified as ‘mild’, there is an increasing demand for combat helmets to protect against these exposures.”

Q: Did the helmets lose protective capacity with repeated blast exposure?

A: There was no trend of decreasing protection with repeated exposure. At the highest exposure level, two of the helmets deformed as addressed in the results section. Added “For all other exposures, helmets experienced minimal deformation and no evidence of degradation by blast exposure was observed.”

Q: The underwash argument sounds very similar to the explanation some unformed individuals give for why seatbelts should not be worn--i.e. if I wear a seatbelt and am involved in a motor vehicle crash, I could be trapped in the wreckage. This has been shown to be patently false. Were you able to assess the validity of this underwash argument? Or did your methods preclude such an assessment. If it is the former, please enumerate. If the latter, consider deleting mention of "underwash."

A: The focus on overhead blast in this study does not allow us to make a direct assessment on the validity of potential blast ‘underwash’. Section altered to “In other orientations, blast wave measurements are complicated by the difference between reflective (measured with pressure gauges oriented parallel to the direction of the blast) and incident (measured with pressure gauges oriented perpendicular to the direction of the blast) pressures, leading to conflicted reports of helmets possibly increasing the risk of primary blast injury [31-34, 41-43]. This risk has to be carefully evaluated, because reflected pressure measurements can be two to eight times greater than incident pressure measurements for the same blast scenario [37].”

Q: I disagree with the first sentence in your conclusions based on your introduction. It seems that primary blast protection was at least part of the motivation for designing helmets, at least in World War I. This may be less true today. Please clarify.

A: Primary blast refers to damage caused by the shock wave resulting from the blast. Secondary blast describes injuries caused by projectiles energized by the blast such as shell fragments. Historical sources such as Dean 1920 provide shrapnel as the main threat addressed by the steel helmets, with some limited ballistic protection. Primary blast effects were poorly understood during World War I, and thus primary blast protection was not a design element.

Added clarification for the distinction between primary and secondary blast in introduction. 

Q: Avoid the use of contractions (e.g. lines 433 and 513).

A: Corrected.

Q: Sentence 425 does not make sense. "Helmets provided...."

A: Corrected sentence order.

Q: Consider deleting the preamble "The authors want to emphasize...." in sentence 428.

A: Removed.

---

## [Decision Letter · Decision Letter 1]

24 Jan 2020

Primary blast wave protection in combat helmet design: A historical comparison between present day and World War I

PONE-D-19-23509R1

Dear Dr. Op 't Eynde,

We are pleased to inform you that your manuscript has been judged scientifically suitable for publication and will be formally accepted for publication once it complies with all outstanding technical requirements.

With kind regards,

David Zonies

Academic Editor

PLOS ONE

Additional Editor Comments (optional):

Reviewers' comments:

Reviewer's Responses to Questions

**Comments to the Author**

1. If the authors have adequately addressed your comments raised in a previous round of review and you feel that this manuscript is now acceptable for publication, you may indicate that here to bypass the “Comments to the Author” section, enter your conflict of interest statement in the “Confidential to Editor” section, and submit your "Accept" recommendation.

Reviewer #1: All comments have been addressed

Reviewer #2: All comments have been addressed

2. Is the manuscript technically sound, and do the data support the conclusions?

Reviewer #1: Yes

Reviewer #2: (No Response)

3. Has the statistical analysis been performed appropriately and rigorously? 

Reviewer #1: Yes

Reviewer #2: (No Response)

4. Have the authors made all data underlying the findings in their manuscript fully available?

Reviewer #1: Yes

Reviewer #2: (No Response)

5. Is the manuscript presented in an intelligible fashion and written in standard English?

Reviewer #1: Yes

Reviewer #2: (No Response)

6. Review Comments to the Author

Reviewer #1: I had accepted the paper during the previous review. I don't have any comments after this revision.

Reviewer #2: (No Response)

7. PLOS authors have the option to publish the peer review history of their article (what does this mean?). If published, this will include your full peer review and any attached files.

Reviewer #1: Yes: Nicholas Jaszczak

Reviewer #2: Yes: Jeremy W. Cannon, MD, SM, FACS

---

## [Editor Report · Acceptance letter]

31 Jan 2020

PONE-D-19-23509R1 

Primary blast wave protection in combat helmet design: a historical comparison between present day and World War I 

Dear Dr. Op 't Eynde:

I am pleased to inform you that your manuscript has been deemed suitable for publication in PLOS ONE. Congratulations! Your manuscript is now with our production department. 

With kind regards,

on behalf of

Dr. David Zonies 

Academic Editor

PLOS ONE